# Development of a New Ramus Anterior Vertical Reference Line for the Evaluation of Skeletal and Dental Changes as a Decision Aid for the Treatment of Crowding in the Lower Jaw: Extraction vs. Nonextraction

**DOI:** 10.3390/jcm14092884

**Published:** 2025-04-22

**Authors:** Ulrich Longerich, Adriano Crismani, Alexandra Mayr, Benjamin Walch, Andreas Kolk

**Affiliations:** 1Academy for Virtual Planning, Orthodontic and Surgical Treatment of Facial Deformities, Karlsplatz 11, D-80335 Munich, Germany; longerich@medicalcube.de; 2Department of Orthodontics, Medical University of Innsbruck, A-6020 Innsbruck, Austria; adriano.crismani@i-med.ac.at; 3Department of Orthodontics, LT Health Care Center, Karlsplatz 11, D-80335 Munich, Germany; a.em.mayr@gmx.de; 4Department of Oral and Maxillofacial Surgery, Medical University of Innsbruck, A-6020 Innsbruck, Austria

**Keywords:** pterygoid vertical (PTV), ramus anterior vertical (RaV), extraction, nonextraction, mandibular crowding, mandibular retromolar region, mandibular dentition growth, Little Irregularity Index, decision-making extraction vs. nonextraction, growth-related development of the posterior molar field

## Abstract

**Objectives:** Anterior crowding in the lower jaw is a common orthodontic issue often managed through premolar extraction, which can affect facial profile development. This study aimed to evaluate skeletal and dental changes in moderate to severe crowding using a novel mandibular reference line—the Ramus Anterior Vertical (RaV)—to support treatment planning. **Methods:** A total of 140 patients (LII > 4 mm and < 9 mm; mean age ≈ 12.5 years) were divided into two groups (G1: extraction; G2: nonextraction; total *n* = 140; *n* = 70 per group). Skeletal and dental parameters were measured before (T0) and after (T1) orthodontic treatment using 280 lateral cephalograms. RaV was defined as a vertical line through the anterior ramus point, perpendicular to the occlusal plane. **Results:** Sagittal measurements relative to RaV were reproducible and unaffected by mandibular mobility. Significant vertical skeletal changes were observed in G2 females, with an increased anterior facial height (N–Sp′ and Sp′–Gn) but a stable Hasund Index. In G1, the dental arch length and distances from RaV to i5 and i6 were reduced, while second molars (i7) remained stable. Sagittal incisor axis changes (L1–NB°, SAi1°) and skeletal–dental correlations (ML–NSL, Gn–tGo–Ar) were present only in G1. **Conclusions:** RaV proved to be a stable mandibular reference for assessing treatment effects. In this study, premolar extraction vs. nonextraction was comparably effective, though some vertical skeletal adaptations, especially in G2 females, took place.

## 1. Introduction

Crowding in the anterior mandibular arch is a common tooth position anomaly in humans, including all sagittal tooth forms, i.e., Angle Classes I, II, and III [1]. Therapeutic options for managing anterior mandibular crowding are limited. Orthodontic arch expansion is possible but constrained by anatomical boundaries. Alternatively, space can be gained through minor interproximal enamel reduction or premolar extraction, the latter of which may significantly impact the facial profile [2,3,4,5]. Notably, in the study of Konstantonis et al. [6], 26.8% of patients underwent treatment involving the extraction of all four first premolars.

In moderate to severe anterior mandibular crowding cases, the decision to extract is not always straightforward. Key factors such as facial profile shape, nasal and lip position, anterior overbite depth, and the dimensions of the apical bases must be considered [7,8]. However, extractions do not guarantee long-term stability, as relapse rates are similar between extraction (Ex) and nonextraction (Non-Ex) treatments [9,10]. In borderline cases, similar to treatment considerations in the upper jaw [11,12], additional decision-making criteria should be taken into account, including the position of the molars, premolars, and incisors when weighing the potential consequences of extraction vs. nonextraction therapy. Specifically, the position of these teeth is crucial in determining whether space can be gained through arch expansion or interproximal enamel reduction instead of extractions. For example, if the molars are significantly mesially inclined, their distalization may create sufficient space without the need for premolar removal. If the incisors are already positioned at a favorable inclination, further proclination to create space may not be advisable, making extraction a more viable option. These factors must be carefully assessed to ensure an optimal balance between functional occlusion, facial profile aesthetics, and long-term stability.

When assessing mandibular dental changes, standard Pterygoid Vertical (PTV), a vertical line drawn through the distal radiographic outline of the pterygoid fissure to the Frankfort horizontal plane, does not allow precise interindividual comparisons in terms of space requirement in the mandible. Unlike the upper dental arch, the lower jaw is uniquely connected to the temporal bone by two double-chamber rotary joints, supported by a masticatory muscle cuff and three strong ligaments. This anatomical structure allows complex combinations of hinge, translational, and grinding movements of the mandible.

The position of the lower jaw relative to the upper jaw is maintained in the habitual occlusion by this intricate functional system, which is richly innervated with proprioceptive sensors in the muscles and nociceptors in the jawbone and teeth, finely regulating the chewing process. However, in many cases, the reproducibility of the mandibular positioning relative to the maxilla can vary significantly between individuals and over time. This variability is often related to functional disturbances of the temporomandibular joint (TMJ), such as articular disc displacement, joint hypermobility, or ligamentous laxity. As discussed by Okeson, changes in the condylar seating caused by alterations in joint structure or position can lead to inconsistent mandibular reference positions during functional movements or radiographic examinations. Such TMJ-related factors contribute to the challenge of achieving consistent mandibular positioning across time points, highlighting the need for more stable skeletal reference structures in orthodontic assessment [13]. Similarly, muscular asymmetries or postural adaptations, such as a forward head posture, have been shown to influence mandibular rest position and dynamic occlusion [14,15]. Parafunctional activities like bruxism or clenching further increase the inconsistency in mandibular positioning by inducing hypertrophy or fatigue of the masticatory muscles, which may alter condylar seating [16]. These factors reduce the reliability of mandibular-based reference lines during orthodontic or cephalometric analysis and highlight the need for more stable intraosseous landmarks in skeletal assessments.

To address these challenges, this study introduced a new static vertical reference line—the Ramus Anterior Vertical (RaV)—allowing precise measurements without interference from mandibular mobility. The inner mandibular region, specifically the newly defined RaV, serves as the mandibular counterpart to the PTV in the maxilla. It is determined by the intersection of the landmark Ra, located at the inner mandibular angle, and the occlusal plane (Figure 1). Extensive literature describes this anatomical region as stable during growth, making it a reliable reference point for mandibular measurements. Enlow and Hans [17] described how the mandible undergoes adaptive changes influenced by biomechanical and functional factors. However, the gonial angle remains stable due to compensatory growth mechanisms, which help maintain balance despite changes in nasal growth, pharyngeal expansion, and basicranial modifications [18]. Further studies by Björk et al. [19,20], Ricketts [21], and Buschang et al. [22] confirmed that the inner mandibular angle remains relatively stable over time, reinforcing RaV’s reliability as a sagittal reference for mandibular assessments.

This study aims to evaluate RaV as an anatomical boundary that defines the available space for mandibular dentition growth in the retromolar region. The goal is to determine whether RaV can serve as an additional tool in therapeutic decision-making between extraction and nonextraction treatment in borderline cases.

## 2. Methods

### 2.1. Study Design and Patient Selection

For this retrospective study, a total of 140 anonymized patients were selected for detailed analysis from an initial population of 1251 orthodontically treated patients with a Little Irregularity Index (LII) between 4 mm and 9 mm. The patients were recruited from two orthodontic centers in Bavaria, Germany, both of which followed a strict standard treatment protocol. This patient cohort was part of a long-term investigation, and all personal data were anonymized prior to analysis, which was conducted at the Department of Oral and Maxillofacial Surgery, Klinikum rechts der Isar. The initial patient cohort comprised two treatment modalities: Group 1 (G1): Patients treated with extraction therapy, both female (F) and male (M) patients (*n* = 625; G1-F: *n* = 338, G1-M: *n* = 287). Group 2 (G2): Patients treated with nonextraction therapy (*n* = 626; G2-F: *n* = 363, G2-M: *n* = 263).

To ensure a scientifically valid, representative, and unbiased sample selection, a two-step stratified random sampling procedure was applied.

In the first step, a randomized, stratified sample of 300 patients was drawn from the original population of 1251 patients, with proportions maintained according to gender and treatment modality. This initial step aimed to preserve population-level representativeness.In the second step, all 300 selected patients were re-evaluated to verify their documented LII values. This was necessary because the initial LII measurements were recorded in the analog patient documentation, which introduced potential for measurement or transcription inaccuracies. Only patients with a confirmed LII between 4 mm and 9 mm were retained. From this validated subsample, a second stratified randomization assigned 35 female and 35 male patients to each group (G1 and G2), resulting in a final study population of 140 patients (G1-F, G1-M, G2-F, G2-M; each *n* = 35).

The rationale for this two-step approach was to prevent inclusion bias from potentially inaccurate documentation and to ensure strict adherence to inclusion criteria. This design is consistent with methodological recommendations for stratified sampling in retrospective observational research [23,24]. A reproducible seed function in R (Version 3.4.1, R Foundation for Statistical Computing, Vienna, Austria) was used to ensure transparency and reproducibility of the sampling process.

### 2.2. Clinical Decision-Making and Justification of Extraction

As no universally fixed criteria exist for extractions in cases with an LII of 4–9 mm, individual treatment decisions were made based on the clinical judgment of the treating orthodontist. Studies have shown that extraction decisions in moderate to severe crowding cases rely on individualized assessments rather than standardized guidelines [25,26,27,28]. Baumrind et al. [27] analyzed extraction decision-making among orthodontists, revealing variability in clinical choices. Keeling et al. [28] highlighted diagnostic imprecision, reinforcing the clinician-dependent nature of extraction decisions. Barber et al. [26] reviewed Class II Division 1 cases, demonstrating that extractions are influenced by specific clinical presentations rather than rigid protocols. Alam [25] reported on severe crowding cases treated with premolar extractions, showing that these decisions are made case by case. However, selection bias was minimized by random selection from a large cohort (*n* = 1251). Additionally, all patients were treated following a strict, uniform protocol, ensuring comparability across both centers.

### 2.3. Group Comparability at Baseline

To validate the group comparability at baseline (T0), statistical analyses assessed skeletal and dental characteristics before treatment. A Welch’s *t*-test for independent samples confirmed no significant baseline differences between Ex and Non-Ex groups (*p* > 0.05 for all numerical variables). This ensures that both groups started from statistically comparable conditions, eliminating concerns regarding pre-existing differences affecting outcomes. Any significant differences observed in skeletal and dentoalveolar measurements (*p* < 0.05) in the analysis of T1–T0 changes can therefore be attributed to treatment modalities rather than baseline discrepancies.

### 2.4. Inclusion and Exclusion Criteria

Inclusion criteria included moderate to severe mandibular crowding (≥4 mm and ≤9 mm) based on Little’s Irregularity Index (LII) [9], treatment with fixed appliances, and the availability of plaster casts and lateral cephalograms (LCs) from T0 (pre-treatment) and T1 (post-treatment). Exclusion criteria comprised adult patients, maxillofacial surgery, previous interproximal enamel reduction (stripping), extraction of teeth other than premolars, periodontal disease (stage II or higher, per AAP & EFP classification) [29], systemic disease (ASA III or higher, per ASA Physical Status Classification) [30], and LCs with geometric aberrations or motion artifacts.

### 2.5. Sample Distribution and Demographics

From the initial 300 patients with LII ≥ 4 mm and ≤ 9, a total of 140 patients were selected. To ensure a balanced distribution, 35 female and 35 male patients were assigned to each subgroup, either treated with extraction (G1) or without extraction (G2), resulting in a total of 140 participants included in the study (Table 1).

The mean age of the female patients was 12.57 ± 2.38 years (median, 12.13), while the mean age of the male patients was 12.41 ± 2.24 years (median, 12.08). The treatment duration was 37.85 ± 11.66 months (median: 37.47) in G1 and 38.08 ± 10.11 months (median: 36.19) in G2.

### 2.6. Cephalometric Imaging and Measurement Protocol

All patients were treated by different orthodontists following a strict standard protocol at both centers. The digital lateral cephalograms (LC) taken at T0 (pre-treatment) and T1 (post-treatment) were analyzed using Ivoris software (Version 8.2.56.110, Computer Konkrete AG, Falkenstein, Germany), and all LC measurements were obtained by the same examiner.

The comparability of Ex and Non-Ex groups at baseline (T0) was statistically validated using Welch’s *t*-test, confirming no significant differences in skeletal and dental parameters (Appendix A, *p* > 0.05 for all variables). This ensures that any observed differences between T0 and T1 result from the respective treatment modalities rather than pre-existing discrepancies (Table 2 and Table 3).

### 2.7. Method Error and Measurement Reliability

To determine the method error, all LCs were measured twice, and additionally, 20 randomly selected LCs were measured by another orthodontist within a week. The average method error was calculated according to Dahlberg [31], and the average reliability coefficient was calculated following Houston [32].

A total of 21 variables (8 angular, 12 linear, 1 ratio index) based on 13 cephalometric points and 1 sagittal and 1 vertical reference plane were used to evaluate the skeletal and dentoalveolar changes (Figure 1, Figure 2 and Figure 3).

### 2.8. Cephalometric Parameters and Measurement Protocol

The following skeletal angles and distances in the LCs were used to assess skeletal changes in the mandible: the sella (S), nasion (N), point A (A), point B (B), gnathion (Gn), articular (Ar), spina dash (Sp’), gonion–tangent point (tGo), spina nasalis line (NSL), mandibular plane (MP = tGo − Gn), and NB line. The following skeletal variables related to the mandible were constructed, measured, and statistically evaluated at T0 and T1 (Figure 2):Skeletal angles: SNB, ANB, SNPg, ML–NSL, ML–NL, Gn–tGo–Ar, and Norderval angle;Linear measurements: anterior facial height (N–Sp’), inferior facial height (Sp’–Gn);Indices: Hasund index.The facial type was classified based on the SNA angle as follows:Retrognathic: SNA < 77°;Orthognathic: 77° ≤ SNA ≤ 85°;Prognathic: SNA > 85°.

The Hasund Index [33] provides an assessment of the anterior vertical facial height (AVFH), categorized as:O (open relation): H index < 71%);N (neutral relation): 71% ≥ H index ≤ 89%);D (deep relation): H index > 89%).

In addition to the Hasund Index, the ML–NL angle was used to evaluate the posterior facial height (PFH) relative to the anterior facial height classification (O, N, or D). Based on Hasund’s digital harmony box and Segner’s tolerance limits, adjusted for individual facial types, the ML–NL angle was classified into three subgroups:Subgroup 1: Large ML–NL angle;Subgroup 2: Balanced ML–NL angle;Subgroup 3: Small ML–NL angle.

To accurately measure dental changes (T1–T0) on the sagittal plane, we introduced a new mandibular vertical reference line: the RaV, which was designed as a stable alternative to the PTV to prevent inconsistencies in both intraindividual and interindividual comparisons.

The RaV was developed by shifting the occlusal plane (Olp) parallel to the internal anterior ascending ramus point (Ra). Similar to the PTV for maxillary dentition, the RaV serves as an anatomical boundary that defines the available space for mandibular dentition growth in the retromolar region (Figure 1).

Sagittal positional changes in the second molars (i7), first molars (i6), second premolars (i5), and incisors (i1) were analyzed by measuring their distances from the RaV to three key reference points, which were established along the inferior tooth axis:Dental centroid point (CP): Located at the intersection of the distal-to-sagittal and superior-to-inferior axes of the enamel–cementum border of the tooth crown;Centre of resistance (CR): Defined at the trifurcation among molars and at 40% of the alveolar height on the coronal view for second premolars [34];Apex (A): Defined as the midpoint between the roots of the molars and the apex of the second premolars.

Additionally, changes in the mandibular central incisors (L1) were measured using the L1–NB angle, which represents the sagittal position and inclination of the L1 relative to the NB line. Sagittal angular changes in tooth axes were evaluated by measuring the angles between the tooth axes of the inferior teeth (second molar (i7), first molar (i6), second premolar (i5), and incisor (i1)) and the mandibular line (ML).

The axes of the lower central incisors were determined by the points at the incisal edge (lie) and the apex (lia) (Figure 1 and Figure 3). Furthermore, changes in the lower incisors were determined using the L1–NB angle. The vertical distances from the ML to the CP, CR, and point A of inferior teeth i7, i6, i5, and i1 were used to measure the vertical changes in tooth position (Figure 1 and Figure 3).

## 3. Statistical Analysis

All statistical analyses were conducted using the statistical software R (version 4.3.1, R Foundation for Statistical Computing, Vienna, Austria). The applied tests included marginal homogeneity tests for discrete trait differences, Welch’s *t*-test [35] for independent sample comparisons at T0 and T1, paired two-sample *t*-tests for within-group comparisons (T0 vs. T1), and multiple linear regression models with beta regression coefficients to analyze sagittal and vertical skeletal influences. Welch’s *t*-test was specifically chosen due to its robustness against unequal variances and unequal sample sizes, making it more reliable than the classical Student’s *t*-test when homogeneity of variance cannot be assumed [36].

To assess the influence of sagittal and vertical skeletal changes on differences in RaV–CRi6, SAi1°, and L1–NB° at T1 compared to T0, standardized multiple linear regression models with beta regression coefficients were applied. Statistical assumptions and validations were ensured through the Central Limit Theorem (Lindeberg–Lévy) [37] for normal distribution with a total sample size of *n* = 280, the Shapiro–Wilk’s test for normality, the Durbin–Watson’s test for absence of autocorrelation, Cook’s distance for outlier detection, the Variance Inflation Factor (VIF) for multicollinearity assessment, Levene’s test for homoscedasticity, and ANOVA for regression model significance evaluation. A significance level (α) of 5% (*p* ≤ 0.05) and a statistical power (1 − β) of at least 90% were assumed. Since no significance was identified, no *p*-value correction was necessary [36].

## 4. Results

The demographic and clinical characteristics of patients are summarized in Table 1. Welsh’s *t*-test revealed no significant difference in the distribution of sex or age between G1 (Ex group) and G2 (Non-Ex group). Additionally, no significant difference was observed in the treatment duration between G1 and G2 (Table 1). The average method error, calculated using Dahlberg’s formula [31], was 0.79 mm for distance measurements and 1.25° for angular measurements. The Houston reliability coefficient [32] for all variables was 0.93, indicating high measurement reliability.

Table 2 presents the differences (T1–T0) in skeletal and dental variables for males and females in G1 and G2. Each subgroup consisted of 35 patients, totaling 140 participants. Welch’s *t*-test showed no significant skeletal changes between T0 and T1 within G1 or G2, except for lower facial height (LFH) and upper facial height (UFH) in G2 (Figure 2, Table 2).

The analysis of 280 LC (T1–T0) stratified by sex revealed significant changes in G2 females for the Norderval angle and the facial heights, but no alterations in the Hasund Index. No significant skeletal changes were obtained between G1 and G2 for other variables. Regarding the sagittal tooth axes, only L1–NB° and SAi1° showed significant differences between G1 and G2 across sexes. The sagittal distance from the RaV to the tooth points was significantly different for the first molars (i6) and second premolars (i5), but not for second molars (i7). In females in G2, RaV–Cri6 increased.

The anterior dental arch length, measured from the molar crowns to the incisal edge (CPi6–lie) and from the molar apices to the incisal apex (Ai6–lia), decreased significantly in G1 compared to G2 in both sexes. In the Ex group (G1), the dental distance from RaV to the crown (CP), center of resistance (CR), and apex (A) of the first molars (i6) and second premolars (i5) significantly decreased in both sexes. However, for the second molars (i7), no significant changes were observed in either sex.

No significant vertical changes (MP to CR of i5, i6, i7, or MP to lie) were observed between T0 and T1 across groups or sexes (*p* > 0.05) (Table 3).

Additional skeletal parameters were assessed as an optional analysis (Appendix A) to evaluate changes in facial type, as described by Björk [38]. In the extraction group (G1), facial type assessment before and after orthodontic treatment in 70 female and male patients showed that 12 patients initially had a retrognathic facial type, 49 were classified as orthognathic, and 9 presented a prognathic facial type. A total of 24 out of 70 patients experienced a change in facial type, with transitions occurring primarily from retrognathic to orthognathic in two cases, from orthognathic to retrognathic in six cases, from orthognathic to prognathic in six cases, and from prognathic to orthognathic in two cases (Appendix A, facial type column, Appendix A). In the nonextraction group (G2), similar changes were observed. Of the eight patients initially classified as retrognathic, five exhibited a transition to an orthognathic facial type. Among the 48 patients with an orthognathic facial type, three developed a retrognathic profile, while eight shifted toward a prognathic classification. Additionally, 2 of 14 patients initially classified as prognathic transitioned to an orthognathic facial type (Appendix A, facial type column, Appendix A).

The comparison of changes in facial type, anterior vertical facial height (AVFH), and posterior facial height (PFH) based on the individualized cephalometric analysis by Hasund and Segner [33] revealed distinct patterns between G1 and G2 from the beginning (T0) to the end (T1) of orthodontic treatment.

In G1, no significant changes in AVFH or PFH were observed in most patients, and facial types remained largely stable. Minor variations in ML–NL angles were noted, reflecting slightly larger angles. Significant changes in PFH were only in patients whose facial type transitioned from prognathic to orthognathic.

In G2, the results followed a similar trend, with the majority of patients showing no significant changes in AVFH or PFH between T0 and T1. However, in the rare cases where facial type shifted from orthognathic to prognathic, PFH was affected.

Regression analysis showed that in G1, ML–NSL (*p* = 0.0037) and Gn–tGo–Ar (*p* = 0.0062) significantly influenced RAV–CRi6 (vertical anterior height change). No significant skeletal predictors were found in G2 (Appendix A). These findings suggest ML–NSL and Gn–tGo–Ar angles influence vertical changes in extraction cases (G1), while skeletal–dental correlations were absent in G2.

## 5. Discussion

To date, no evidence-based guidelines exist for determining the necessity of premolar extraction in patients with moderate anterior crowding of the lower jaw. The known side effects of nonextraction therapy must always be weighed against the potential positive and negative facial consequences of premolar extraction [2,3,4,5,7,8,38,39]. Therefore, this study aimed to examine the skeletal effects of premolar extraction (G1-F: 35, G1-M: 35, *n* = 70) compared to nonextraction therapy (G2-F: 35, G2-M; 35, *n* = 70) in moderate to severe mandibular anterior crowding using a newly introduced sagittal reference line (RaV).

The stratified random sampling approach ensured a representative sample, preserving gender and treatment group proportions while preventing selection bias. The reproducible seed function further strengthened methodological transparency and replicability, enhancing the internal validity and generalizability of the findings within the defined inclusion criteria.

This study reflects a real-world clinical setting, where extraction decisions in moderate to severe crowding (LII: 4–9 mm) are made individually, aligning with previous findings that emphasize clinician-dependent treatment planning rather than fixed guidelines [25,26,27,28]. The randomization process and balanced group distribution further ensured unbiased comparisons, reinforcing the conclusion that nonextraction therapy is as effective as extraction therapy in managing moderate to severe crowding when vertical skeletal dimensions permit.

The baseline comparability of extraction and nonextraction groups was statistically validated using a Welch’s *t*-test, confirming no significant pre-treatment differences in skeletal and dental parameters (Appendix A, *p* > 0.05 for all numerical variables). As detailed in the methods section, this ensures that any observed differences between T0 and T1 are attributable to the respective treatment modalities rather than pre-existing skeletal and dental variations (Table 2 and Table 3).

A key contribution of this study is the introduction of the new reference line RaV, serving as the mandibular counterpart to the PTV reference line in the maxilla. Unlike the latter, which is influenced by mandibular mobility [13,14,15,16,40], the RaV provides a stable reference, eliminating positional variability and enabling precise sagittal comparisons of dental variables. In this study, the use of the RaV represents a significant methodological advancement in the assessment of sagittal dental changes, particularly for evaluating mandibular molar distalization. Compared to traditional reference lines, such as the PTV or occlusal plane-based references, the RaV offers enhanced intraindividual and interindividual reproducibility by anchoring the reference point to a stable mandibular structure. While other methods, such as using NB or mandibular plane lines, are commonly employed, they often lack the same degree of skeletal stability due to variation in mandibular posture or underlying dental compensation. Studies relying on these conventional lines are prone to error when interpreting treatment-related dental movements, especially in long-term comparisons [41,42]. Thus, RaV provides a more reliable and anatomically justified alternative. However, further studies comparing RaV to 3D superimposition techniques or cone beam-based skeletal analysis would be necessary to confirm its superiority across broader applications.

This raises the question: Are other available methods equally scientifically efficient in this assessment? While widely used, they may not offer the same skeletal anchorage or reproducibility, and their performance should be directly compared to RaV in future research. Although alternative approaches such as PTV, NB, or mandibular plane-based references are commonly employed, their scientific efficiency, particularly regarding long-term reproducibility and anatomical stability, remains a topic for future systematic evaluation.

Despite the variation in anterior crowding severity, skeletal analysis showed only minor significant differences between the extraction and nonextraction groups, irrespective of gender, with the exception of female patients in the nonextraction group G2 (Table 3). In this group, a significant increase in anterior upper and lower facial height (N–Sp’ and Sp’–Gn) was observed without affecting the Hasund Index (Table 2), whereas no comparable changes were observed in the extraction group (G1). These findings contradict several previous studies reporting a significant reduction in vertical dimensions after premolar extraction [43,44,45,46,47,48,49,50,51]. However, in G1, a smaller reduction in anterior upper facial height was observed, suggesting that extraction in G1 had only a marginal influence on vertical dimensions. Our results partially confirm the studies by Kim et al. [45] and Aras [52], who also reported a nonsignificant reduction in the ML–NSL angle in open bite–bite patients following extraction. While Kocadereli et al. [46] reported a significant increase in the ML–NSL angle in nonextraction patients, our findings showed a non-significant change in this value. This suggests that the impact of extraction therapy on vertical skeletal relationships may be minor and subject to individual variation, e.g., the initial amount of crowding.

Regarding facial type changes, as described by Björk [38], 23% of patients (16 out of 70) in G1 experienced changes, whereas 18 (26%) patients in G2 showed a change in facial type, each in connection with a stable mean Hasund/Segner Index.

The observation that only two patients in G1 showed a shift in facial type from prognathic to orthognathic, accompanied by a PFH balance change from N1 to N2, is of particular clinical interest (Appendix A). This very limited occurrence underscores the exceptional stability of PFH following premolar extraction. It aligns closely with the findings of Kirschneck et al. [53], who found no significant sagittal or vertical skeletal changes attributable to extractions. In contrast to other studies [51,52] that have suggested a broader skeletal impact from extractions, our result strengthens the evidence that such changes are not only rare but also likely incidental rather than treatment-induced. This further confirms that skeletal dimensions, especially posterior facial height, remain largely unaffected by premolar extraction therapy in the context of moderate to severe mandibular crowding.

The use of the RaV reference allowed for the elimination of measurement errors caused by mandibular mobility, which was crucial for ensuring comparability of potential sagittal tooth position changes. Our results showed a significant increase in the sagittal distance between the RaV reference line and i5, i6, while no significant changes were observed for i7. This suggests a growth-related development of the posterior molar field during the first mixed dentition phase, a phenomenon that has rarely been described. These findings are consistent with studies by Kim et al. [54], Turkoz et al. [55], and Richardson [56].

As expected, orthodontic treatment (OT) in G2 resulted in fewer changes in dental variables, and the sagittal distances between i5, i6, and i7 and the RaV remained stable [57,58,59]. However, in both G1 and G2, a significant reduction in CPi6–lie and Ai6–lia was observed, contradicting Kumari et al. [47], who reported no such reduction in nonextraction patients. Liu et al. [60] found that nonextraction therapy resulted in increased lingual inclination of mandibular incisors, which may explain the reduction in sagittal arch length observed in this study. These findings were further confirmed by significant alterations in SA–i7, SA–i6, SA–i5, and SA–i1 axes between G1 and G2, independent of gender (Table 3).

A significant difference in the mean vertical distance from the mandibular plane (MP) to the center of resistance (CR) of i7, i6, and i5 and between T0 and T1 was only observed in females, consistent with previous studies reporting no significant vertical changes between extraction and nonextraction patients [47]. Further pre/post-OT analyses revealed no significant morphological differences between G1 and G2.

This study demonstrates that sagittal and vertical facial structures can be slightly influenced by premolar extraction, contrasting with Villard and Patcas [61], who suggested that moderate anterior crowding cases exhibit significant skeletal differences depending on treatment choice. Further prospective studies are needed to better understand these effects.

The clinical implications of the findings of this study are substantial. While molar distalization using skeletal anchorage is well-established in the maxilla (e.g., Beneslider [57]), this process is more complex in the mandible due to anatomical constraints. The RaV reference line provides a precise measurement tool for planning mandibular molar distalization, offering an alternative to PTV for accurate space assessment before treatment.

A preliminary study demonstrated that PTV serves as a reliable anatomical reference line for estimating the achievable distalization distances before initiating OT [11]. However, crowding in the mandible presents an even greater challenge, as molar distalization is inherently more complex due to considerable anatomical limitations compared to the maxilla. In some cases, it may not be feasible at all.

The RaV reference line offers a valuable alternative, analogous to the PTV, allowing for more precise evaluation of space availability before treatment planning. Accurately determining potential movement distances allows for better OT planning, improving space creation in borderline cases of moderate crowding. In some instances, this may even eliminate the need for extractions or aesthetic compromises compared to initial clinical assessments.

By accurately determining potential movement distances, OT planning can be significantly optimized, increasing the likelihood of successful space creation in borderline cases of moderate crowding. In some instances, this may even eliminate the need for extractions or aesthetic compromises compared to initial treatment plans based on clinical assessment. By implementing the new reference line, this study is the first to provide evidence of its clinical significance.

However, certain limitations must be acknowledged. These include the retrospective study design, the heterogeneous range of anterior crowding, variability in biological growth ages, and the limited significance due to the variety of parameters examined. Consequently, the findings should be interpreted as preliminary, requiring further clinical validation to assess the applicability and effectiveness of the RaV reference line as a supportive tool in mandibular orthodontic treatment planning. Due to the retrospective nature of this study, causal conclusions about the final clinical value of the RaV reference line cannot be drawn. Although the sample size (*n* = 140) was statistically sufficient for analyzing normally distributed data, a larger sample size would improve the statistical power, particularly for detecting subtle skeletal effects. A stratified analysis based on crowding severity could further refine treatment recommendations. Although Welch’s *t*-test was applied to account for variance heterogeneity, further prospective studies with more controlled research questions are needed to establish robust clinical guidelines.

This study supports the RaV as a stable and reproducible reference line for assessing mandibular dental changes. While skeletal changes were minimal, particularly in the posterior facial height, dental parameters such as arch length and incisor inclination showed notable treatment effects, especially in the extraction group. These findings offer valuable input for individualized treatment planning in orthodontics. This has clinical implications for orthodontic treatment planning. For an expanded evaluation of the clinical applicability of the RaV reference line, additional prospective studies should be conducted, particularly focusing on comparative studies between the RaV and PTV reference lines to validate measurement accuracy. Long-term studies should also assess the stability of mandibular molar distalization, especially in combination with skeletal anchorage techniques.

## 6. Conclusions

This study introduced the RaV as a new skeletal reference plane for evaluating mandibular dental changes and evaluated its clinical impact in patients with moderate to severe anterior crowding after premolar extraction vs. nonextraction treatment. Multiple regression analysis identified ML–NSL and Gn–tGo–Ar as significant skeletal predictors of RaV–CRi6 changes in the extraction group, while no such associations were found in the nonextraction group. Dental effects such as changes in L1–NB° and SA–i1° were more pronounced in extraction cases. Most patients retained their facial type, with only minor transitions observed. Vertical skeletal changes were absent in males but evident in G2 females, suggesting possible treatment-related vertical adaptations. While the new RaV proves promise as an adjunctive tool in orthodontic diagnostics, further prospective studies are needed to confirm these findings under controlled conditions.

## Figures and Tables

**Figure 1 jcm-14-02884-f001:**
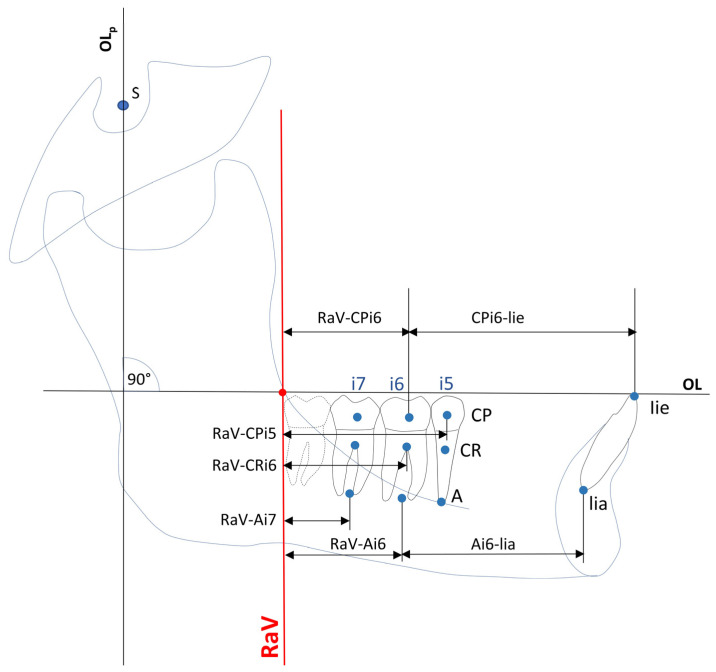
Construction of sagittal dental variables. Abbreviations: CP = dental centroid point; CR = dental center of resistance; A = dental apex point; i5/i6/i7 = second premolar, second and first molars; lie = lower incisor edge; lia = lower incisor apex; OLp = perpendicular line to occlusion line through sella of the Panscherz analysis; OL = occlusion line; RaV = anterior ramus vertical; RaV–CPi5/i6/i7 = sagittal distances between RaV and centroid point of second premolar, second and first molars; RaV–Cri5/i6/i7 = sagittal distances between RaV and center of resistance points of second premolar, second and first molars; RaV–Ai5/i6/i7 = sagittal distances between RaV and apex points of second premolar, second and first molars; CPi6–lie = sagittal distance between dental centroid point and RaV and upper incisor edge; Ai6–lia = sagittal distances between dental apex point i6 and lower incisor apex.

**Figure 2 jcm-14-02884-f002:**
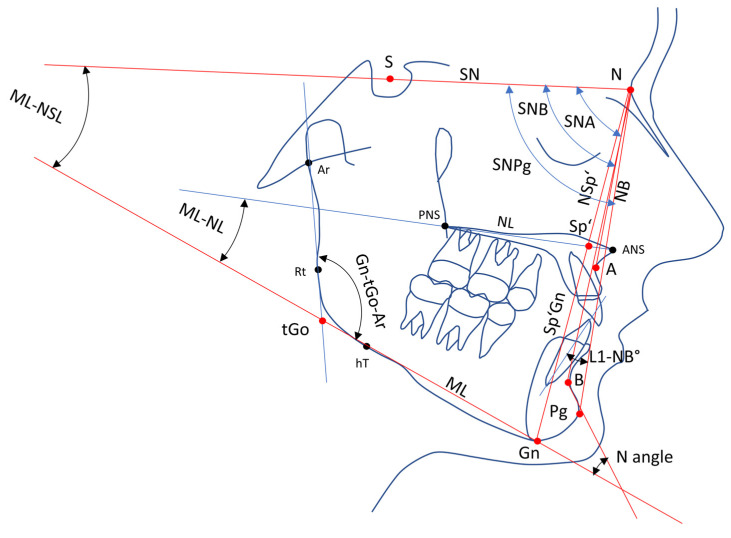
Skeletal variables, sagittal and vertical reference lines. Abbreviations: S = sella point; N = nasion; Ar = articulare; A = deepest point on curvature of maxillary; Sp’ = spina prime; B = most anterior measure point of mandibular apical base; ANS = anterior nasal spine; PNS = posterior nasal spine; Pg = pogonion; Gn = gnathion; tGo = gonion tangent point; hT = posterior construction point of mandibular line; Rt = caudal construction point of ramus line; SN = sella–nasion line; NL = nasale line; ML = mandibular line; NB = nasal–B line; SNA = angle between sella–nasale line and nasale–A line; SNB = angle between sella–nasale line and nasale–B line; SNPg = angle between sella–nasale line and nasale–pogonion line; Gn–tGo–Ar = angle between ramus line and mandibular line; ML–NSL = inclination of mandibula to sella–nasale line; ML–NL = angle between nasal line and mandibular line; L1–NB = the angle between long axis of the mandibular central incisor and nasion–point B line; N–Sp’ = upper facial height; Sp’–Gn = lower facial height.

**Figure 3 jcm-14-02884-f003:**
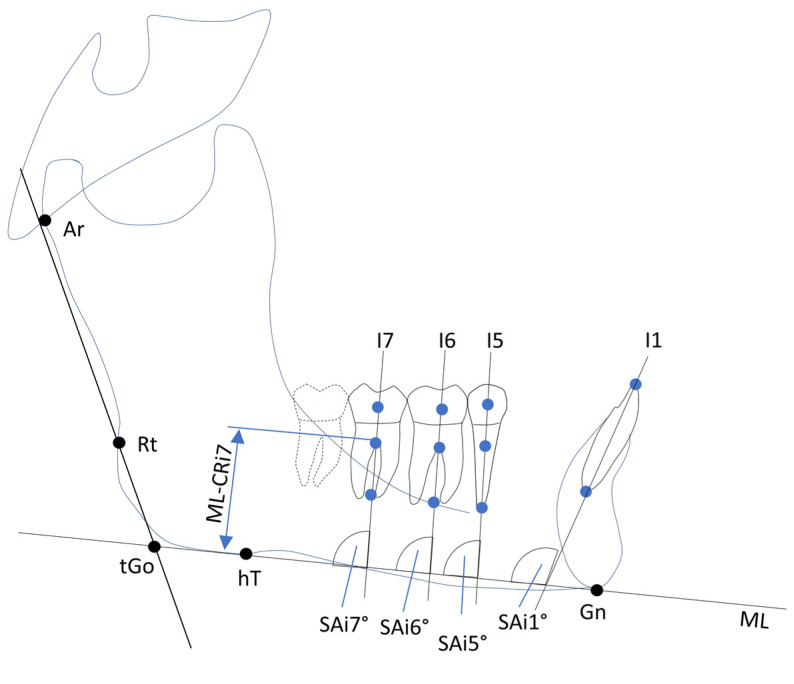
Construction of vertical dental variables and teeth inclinations. Abbreviations: i1/i5/i6/i7 = lower incisor, second premolar, second and first molars; Ar = articulare; Rt = caudal construction point of ramus line; hT = construction point of mandibular line; tGo = angle between ramus line and mandibular line; Gn = gnathion; ML = mandibular line; ML–Cri1/i5/i6/i7 = vertical distance between mandibular line and centers of resistance of lower incisor, second premolar, first and second molars; SAi1/i5/i6/i7 = inclination of lower incisor, second premolar, first and second molars to mandibular line.

**Table 1 jcm-14-02884-t001:** Characteristics of the study population.

Beginning of Treatment (T0)	Total Ex + Non-Ex	Ex	Non-Ex	*p* Value Welch’s *t*-Test H0: Ex = Non-Ex H1: Ex ≠ Non-Ex
Number of patients:				----
All (female + male)	140	140	140	
female	70	70	70	
male	70	70	70	
Mean age (total) Median	12.49 ± 2.30 12.11	12.58 ± 2.34 12.06	12.39 ± 2.27 12.14	*p* = 0.62
Mean age (female) Median	12.57 ± 2.38 12.13	12.55 ± 2.31 12.12	12.59 ± 2.48 12.13	*p* = 0.94
Mean age (male) Median	12.41 ± 2.24 12.08	12.62 ± 2.40 12.06	12.20 ± 2.07 12.15	*p* = 0.43
Treatment duration (T1–T0) Months Median	37.97 ± 10.87 36.55	37.85 ± 11.66 37.47	38.08 ± 10.11 36.19	*p* = 0.90

Abbreviations: H_0_ = null hypothesis; H_1_ = alternative hypothesis.

**Table 2 jcm-14-02884-t002:** Changes in skeletal and dental variables before (T0) and after (T1) treatment in the extraction group with gender subgroups. Abbreviations: H_0_ = null hypothesis; H_1_ = alternative hypothesis; SNA = angle between sella, nasion, and subspinale point A; SNB = angle from sella, nasion, and point of greatest concavity on anterior surface of mandibular symphysis; ANB = angle between A point, nasion, B point; SNPg = angle between sella, nasion, and pogonion; Gn–tGo–Ar = angle between ramus line and mandibular line; NSBa = cranial base flexure angle; NL–NSL = inclination of maxilla; ML–NSL = inclination of mandible; ML–NL = angle between mandibular line and nasion–sella line; N–Sp’ = upper facial height; Sp’–Gn = lower facial height; L1–NB = angle between long axis of mandibular central incisor and nasion–point B line; SAi1/i5/i6/i7 = Inclination of lower incisor, second premolar, first and second molars to mandibular line; RaV–CPi5/i6/i7 = sagittal distances between RaV to centroid point of second premolar, first and second molars; RaV–CRi5/i6/i7 = sagittal distances between RaV to centre of resistance of second premolar, first and second molars; RaV–Ai5/i6/i7 = sagittal distances between RaV to apex point of second premolar, first and second molars; CPi6–lie = sagittal distance between centroid point of first molar to lower incisor edge; Ai6–lia = sagittal distance between apex point of first molar to apex of lower incisor; MP–CRi5/i6/i7 = vertical distances between mandibular line or plane to centre of resistance of second premolar, first and second molars; MP–lie: vertical distances between mandibular line or plane to lower incisor edge.

Skeletal and Dental Variables	Ex Group Differences (T1–T0)		Welch’s *t*-Test	Non-Ex Group Differences (T1–T0)		Welch’s *t*-Test
			H0: ♀ = ♂			H0: ♀ = ♂
			H1: ♀ ≠ ♂			H1: ♀ ≠ ♂
	Female	Male	*p* Value	Female	Male	*p* Value
SNA°	−0.32 ± 3.28	−0.55 ± 2.81	0.7492	0.61 ± 2.52	0.21 ± 2.96	0.5421
SNB°	0.39 ± 2.80	0.01 ± 2.24	0.5360	1.18 ± 2.30	0.70 ± 2.60	0.5166
ANB°	−0.66 ± 1.90	−0.57 ± 1.73	0.8287	−0.33 ± 2.02	−0.58 ± 1.94	0.5884
SNPg°	0.99 ± 2.85	0.44 ± 2.45	0.3907	1.39 ± 2.18	1.01 ± 2.76	0.5193
Gn–tGo–Ar°	−1.8 ± 6.71	−1.43 ± 4.8	0.4054	−2.02 ± 4.39	−1.83 ± 6.49	0.8888
NSBa	−0.39 ± 3.3	0.17 ± 4.42	0.5478	−0.03 ± 3.40	−0.46 ± 4.05	0.6331
NL–NSL°	−0.19 ± 3.56	0.19 ± 3.13	0.6392	−0.16 ± 2.41	0.15 ± 3.03	0.9826
ML–NSL°	−1.02 ± 3.33	0.01 ± 3.33	0.2381	−1.79 ± 2.85	−1.01 ± 3.50	0.3070
ML–NL°	0.89 ± 2.94	−0.29 ± 4.15	0.4879	−1.75 ± 2.93	1.27 ± 3.38	0.5271
Norderval°	3.15 ± 3.29	3.06 ± 4.95	0.9233	0.99 ± 3.21	1.91 ± 5.07	0.3698
N–Sp’ mm	−0.45 ± 5.54	0.95 ± 7.71	0.4200	0.69 ± 6.04	−4.77 ± 6.37	** 0.0004 **
Sp’–Gn mm	0.72 ± 6.15	1.10 ± 5.59	0.8524	−3.91 ± 6.99	2.21 ± 7.22	** 0.0005 **
Hasund Index%	−2.45 ± 6.14	−0.74 ± 5.24	0.2144	−2.41 ± 6.05	−2.47 ± 4.27	0.9583
L1–NB°	−2.83 ± 6.83	−3.33 ± 6.62	0.7553	1.23 ± 5.40	0.77 ± 5.96	0.7315
SAi7°	−3.54 ± 5.92	−1.52 ± 7.73	0.2819	−6.33 ± 6.67	−3.03 ± 9.50	0.9529
SAi6°	−2.97 ± 8.05	−2.83 ± 6.54	0.9365	0.93 ± 4.26	−1.63 ± 6.37	0.0526
SAi5°	−1.4 ± 6.93	−1.57 ± 6.91	0.9204	0.53 ± 5.53	−1.59 ± 5.74	0.1212
SAi1°	−1.99 ± 7.63	−3.45 ± 6.39	0.3887	2.11 ± 4.87	0.73 ± 6.17	0.3026
RaV–CPi7 mm	4.83 ± 4.06	4.49 ± 4.43	0.262	1.38 ± 3.8	1.56 ± 4.19	0.8516
RaV–CPi6 mm	3.95 ± 4.55	3.47 ± 4.27	0.6513	1.78 ± 3.52	1.11 ± 5.36	0.5409
RaV–CPi5 mm	3.72 ± 4.61	2.15 ± 6.95	0.2721	0.22 ± 4.46	0.42 ± 6.28	0.8768
RaV–CRi7 mm	1.8 ± 3.41	1.09 ± 4.23	0.4434	0.77 ± 3.51	0.74 ± 3.16	0.9749
RaV–CRi6 mm	4.49 ± 4.06	3.47 ± 4.46	0.3223	2.36 ± 3.60	1.73 ± 5.37	0.5679
RaV–CRi5 mm	4.11 ± 4.33	3.19 ± 5.14	0.4222	0.84 ± 4.19	0.88 ± 6.31	0.9734
RaV–Ai7 mm	0.58 ± 4.19	0.69 ± 3.97	0.9105	0.23 ± 3.16	0.54 ± 3.56	0.7049
RaV–Ai6 mm	4.45 ± 3.91	4.37 ± 4.38	0.9323	1.27 ± 6.00	1.62 ± 5.54	0.7997
RaV–Ai5 mm	4.11 ± 4.55	2.91 ± 5.92	0.3483	0.90 ± 4.74	1.05 ± 6.62	0.9179
CPi6–lie mm	−6.35 ± 4.95	−5.87 ± 6.25	0.7231	−4.26 ± 5.25	−3.06 ± 5.52	0.0954
Ai6–lia mm	−6.26 ± 4.14	−5.83 ± 5.60	0.4898	−4.38 ± 4.9	−2.95 ± 5.60	0.2628
MP–CRi7 mm	2.2 ± 5.19	3.01 ± 4.61	0.4126	2.46 ± 6.19	2.97 ± 4.38	0.1793
MP–CRi6 mm	2.37 ± 3.62	2.56 ± 5.45	0.8590	2.30 ± 2.96	2.24 ± 4.46	0.2573
MP–CRi5 mm	2.70 ± 4.81	2.87 ± 4.81	0.8761	2.17 ± 4.05	2.66 ± 5.12	0.1585
MP–lie mm	−2.39 ± 6.08	−2.72 ± 5.76	0.5250	−0.83 ± 4.54	0.52 ± 4.01	0.4901

**Table 3 jcm-14-02884-t003:** Comparison of the changes in skeletal and dental differences of variables (T1–T0) between extraction and nonextraction groups with gender subgroups. Abbreviations: T0 = begin of orthodontic treatment, T1 = end of orthodontic treatment; Ex = extraction therapy; Non-Ex = nonextraction therapy; H_0_ = null hypothesis; H_1_ = alternative hypothesis; SNA = angle between sella, nasion, and subspinale point A; SNB = angle from sella, nasion, and point of greatest concavity on anterior surface of mandibular symphysis; ANB = angle between A point, nasion, B point; SNPg = angle between sella, nasion, and pogonion; Gn–tGo–Ar = angle between ramus line and mandibular line; NSBa = cranial base flexure angle; NL–NSL = inclination of maxilla; ML–NSL = inclination of mandible; ML–NL = angle between mandibular line and nasion–sella line; N–Sp’ = upper facial height; Sp’–Gn = lower facial height; L1–NB = angle between long axis of mandibular central incisor and nasion–point B line; SAi1/i5/i6/i7 = inclination of lower incisor, second premolar, first and second molars to mandibular line; RaV–CPi5/i6/i7 = sagittal distances between RaV to centroid point of second premolar, first and second molars; RaV–CRi5/i6/i7 = sagittal distances between RaV to centre of resistance of second premolar, first and second molars; RaV–Ai5/i6/i7 = sagittal distances between RaV to apex point of second premolar, first and second molars; CPi6–lie = sagittal distance between centroid point of first molar to lower incisor edge; Ai6–lia = sagittal distance between apex point of first molar to apex of lower incisor; MP–CRi5/i6/i7 = vertical distances between mandibular line or plane to centre of resistant of second premolar, first and second molars; MP–lie: vertical distances between mandibular line or plane to lower incisor edge.

Skeletal and Dental Variables	Female		Welch’s *t*-Test	Male		Welch’s *t*-Test
			H0: Ex = Non-Ex			H0: Ex = Non-Ex
	Ex	Non-ex	H1: Ex ≠ Non-Ex	Ex	Non-ex	H1: Ex ≠ Non-Ex
	T1–T0	T1–T0	*p* Value	T1–T0	T1–T0	*p* Value
SNA°	−0.32 ± 3.28	0.61 ± 2.52	0.1857	−0.55 ± 2.81	0.21 ± 2.96	0.2715
SNB°	0.39 ± 2.80	1.18 ± 2.30	0.2041	0.01 ± 2.24	0.79 ± 2.60	0.1836
ANB°	−0.66 ± 1.90	−0.33 ± 2.02	0.4779	−0.57 ± 1.73	−0.58 ± 1.94	0.9689
SNPg°	0.99 ± 2.85	1.39 ± 2.18	0.5123	0.44 ± 2.45	1.01 ± 2.76	0.3701
Gn–tGo–Ar°	−2.68 ± 7.37	−2.02 ± 5.34	0.6519	−1.43 ± 4.80	−1.83 ± 6.49	0.7687
NSBa	−0.39 ± 3.30	−0.03 ± 3.40	0.6543	0.17 ± 4.42	−0.46 ± 4.05	0.5351
NL–NSL°	−0.19 ± 3.56	−0.16 ± 2.41	0.9718	0.19 ± 3.13	−0.15 ± 3.03	0.6486
ML–NSL°	−1.02 ± 3.88	−1.79 ± 2.85	0.3429	0.01 ± 3.33	−1.01 ± 3.50	0.2162
ML–NL°	−0.89 ± 2.94	−1.75 ± 2.93	0.2259	−0.29 ± 4.15	−1.27 ± 3.38	0.2837
Norderval°	3.15 ± 3.29	0.99 ± 3.21	** 0.0069 **	3.06 ± 4.95	1.91 ± 5.07	0.3414
N–Sp’ mm	−0.45 ± 6.73	−4.77 ± 6.37	** 0.0074 **	0.95 ± 7.7	0.69 ± 6.04	0.8745
Sp’–Gn mm	0.72 ± 8.14	−3.91 ± 6.99	** 0.0128 **	1.1 ± 8.59	2.21 ± 7.22	0.5586
Hasund Index%	−2.45 ± 6.14	−2.41 ± 6.05	0.9750	−0.74 ± 5.24	−2.47 ± 4.27	0.1348
L1–NB°	−2.83 ± 6.83	1.23 ± 5.40	** 0.0075 **	−3.33 ± 6.62	0.77 ± 5.96	** 0.0083 **
SAi7°	−3.83 ± 9.95	−2.89 ± 10.25	0.6966	−1.52 ± 7.73	−3.03 ± 9.50	0.4696
SAi6°	−1.97 ± 8.05	0.93 ± 4.26	0.1435	−2.83 ± 6.54	−1.63 ± 6.37	0.4393
SAi5°	−1.40 ± 6.93	0.53 ± 5.53	0.2030	−1.57 ± 6.9	−1.59 ± 5.74	0.9880
SAi1°	−1.99 ± 7.63	2.11 ± 4.87	** 0.0095 **	−3.45 ± 6.39	0.73 ± 6.17	** 0.0069 **
RaV–CPi7 mm	2.83 ± 4.06	1.38 ± 3.88	0.1319	0.33 ± 5.07	1.56 ± 4.19	0.2721
RaV–CPi6 mm	3.95 ± 4.55	1.78 ± 3.52	** 0.0287 **	3.47 ± 4.27	1.11 ± 5.36	** 0.0455 **
RaV–CPi5 mm	3.72 ± 4.61	0.22 ± 4.46	** 0.0019 **	3.15 ± 7.00	0.42 ± 6.28	** 0.0281 **
RaV–CRi7 mm	1.80 ± 3.41	0.77 ± 3.51	0.2149	1.09 ± 4.23	0.74 ± 3.16	0.6932
RaV–CRi6 mm	4.49 ± 4.06	2.36 ± 3.02	** 0.0233 **	3.47 ± 4.46	0.73 ± 5.37	** 0.0345 **
RaV–CRi5 mm	4.11 ± 4.33	0.84 ± 4.19	** 0.0020 **	3.19 ± 5.14	0.88 ± 6.31	** 0.0982 **
RaV–Ai7 mm	0.58 ± 4.19	0.23 ± 3.16	0.6968	0.69 ± 3.97	0.54 ± 3.56	0.8673
RaV–Ai6 mm	4.37 ± 6.01	1.27 ± 4.38	** 0.0110 **	4.37 ± 4.38	1.62 ± 5.54	** 0.0247 **
RaV–Ai5 mm	4.11 ± 4.55	0.90 ± 4.74	** 0.0052 **	3.91 ± 5.92	1.05 ± 6.62	** 0.0378 **
CPi6–lie mm	−6.35 ± 4.95	−3.26 ± 5.25	** 0.0457 **	−5.87 ± 6.25	−1.67 ± 6.27	** 0.0065 **
Ai6–lia mm	−6.26 ± 4.14	−3.38 ± 4.50	** 0.0421 **	−5.43 ± 5.71	−1.95 ± 5.60	** 0.0143 **
MP–CRi7 mm	1.92 ± 5.19	1.36 ± 6.10	0.2507	2.88 ± 4.54	2.97 ± 4.38	0.9398
MP–CRi6 mm	2.37 ± 3.62	1.30 ± 2.96	** 0.1120 **	2.56 ± 5.45	2.24 ± 4.46	0.7846
MP–CRi5 mm	2.70 ± 4.26	−1.17 ± 4.05	** 0.0052 **	2.87 ± 4.81	2.66 ± 5.12	0.8612
MP–lie mm	−0.39 ± 5.73	−2.53 ±4.85	0.0969	0.55 ± 6.57	0.57 ± 7.71	0.9917

## Data Availability

The data underlying this article will be shared upon reasonable request to the corresponding author.

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
