# Peer review of "Development of a New Ramus Anterior Vertical Reference Line for the Evaluation of Skeletal and Dental Changes as a Decision Aid for the Treatment of Crowding in the Lower Jaw: Extraction vs. Nonextraction"

_jcm, 2025, doi:10.3390/jcm14092884_

Round 1

Reviewer 1 Report (Previous Reviewer 2)

Comments and Suggestions for Authors

The Authors responded to all my previous requests for changes in the text.

However, I think the Abstract is too long and needs to be reduced.

Author Response

The Authors responded to all my previous requests for changes in the text.

However, I think the Abstract is too long and needs to be reduced.

Done, similar recommendation as given by reviewer II, abstract has been significantly reduced

Reviewer 2 Report (Previous Reviewer 3)

Comments and Suggestions for Authors

Dear authors,

thank you for your contribution. I recommend the following revisions.

Abstract:

Please reduce section of Conclusion, it is too long and repetitive. You can also move some content to the final section, Abstract usually requires a few sentences.

Introduction:

-“However, in many cases, the reproducibility of the mandibular positioning relative to the maxilla can vary significantly between individuals and over time.": please add some examples, different from the newly introduced Ramus Anterior Vertical (RaV).

Methods:

-Methodology is accurate, but I suggest to present it in subparagraph to make it fluent.

-"To ensure a scientifically valid, representative, and unbiased sample selection, a two- step stratified random sampling approach was applied.": please specify why you consider a one-step random sampling approach (the first one you mentioned) not sufficient and scientifically valid.

Results:

Please reduce some redundant text.

Discussion

-“This represents a significant methodological advancement, particularly in the assessment of molar distalization and the evaluation of dental changes during treatment in the mandible.": are the other available methods equally scientifically efficient in this assessment? Please discuss.

-“Only in two patients in G1 whose facial type shifted from prognathic to orthognathic, did the posterior facial height (PFH) balance change from N1 to N2 (Table 4 a, supplementary Materials). This aligns with the findings of Kirschneck et al. [37], who also reported no significant effects of premolar extraction on sagittal or vertical skeletal structures.": please better highlight this finding compared to the literature.

Author Response

thank you for your contribution. I recommend the following revisions.

(author´s answers and consecutive overwork are highlighted in red)

Abstract:

Please reduce section of Conclusion, it is too long and repetitive. You can also move some content to the final section, Abstract usually requires a few sentences.

Done, abstract and especially the conclusion section has been shortened. Redundant information has been eliminated.

Introduction:

-“However, in many cases, the reproducibility of the mandibular positioning relative to the maxilla can vary significantly between individuals and over time.": please add some examples, different from the newly introduced Ramus Anterior Vertical (RaV).

Done, examples have been given.

Methods:

-Methodology is accurate, but I suggest to present it in subparagraph to make it fluent.

-"To ensure a scientifically valid, representative, and unbiased sample selection, a two- step stratified random sampling approach was applied.": please specify why you consider a one-step random sampling approach (the first one you mentioned) not sufficient and scientifically valid.

Methodology is now presented as a subparagraph, in addition it has been specified, why the one-step random sampling approach is not sufficient and scientifically valid.

Results:

Please reduce some redundant text.

Done, repetitive passages have been eliminated, the passage has been shortened.

Discussion

-“This represents a significant methodological advancement, particularly in the assessment of molar distalization and the evaluation of dental changes during treatment in the mandible.": are the other available methods equally scientifically efficient in this assessment? Please discuss.

this topic has been added has a further aspect in the discussion section.

-“Only in two patients in G1 whose facial type shifted from prognathic to orthognathic, did the posterior facial height (PFH) balance change from N1 to N2 (Table 4 a, supplementary Materials). This aligns with the findings of Kirschneck et al. [37], who also reported no significant effects of premolar extraction on sagittal or vertical skeletal structures.": please better highlight this finding compared to the literature.

Only in two patients in G1 whose facial type shifted from prognathic……

These findings have been highlighted compared to the data from Kirschneck et al and other authors with similar data in terms of only minor effects of premolar extraction.

This manuscript is a resubmission of an earlier submission. The following is a list of the peer review reports and author responses from that submission.

Round 1

Reviewer 1 Report

Comments and Suggestions for Authors

The aim of the manuscript “Decision Making in Terms of Treatment of Crowding in the Lower Jaw: Extraction vs. Nonextraction - Development of a New Ramus Anterior Vertical Reference Plane for Evaluation of Skeletal and Dental Changes” was to assess the extraction versus non-extraction protocol in patients with mild to moderate anterior mandibular crowding. To facilitate this assessment, a new vertical reference plane in the mandible, the ramus anterior vertical (RaV) plane, was developed.

The manuscript is both interesting and valuable in aiding clinical decision-making. However, before it can be recommended for acceptance, some corrections, improvements, and clarifications are necessary.

Comments:

References: Please revise all references in the manuscript to ensure they are accurate and adhere to the journal’s specified format.

Material and Methods:
First of all, some information is assumed to be known by the reader, and abbreviations are used without prior explanation. For example, PTV plane—please define it when it first appears in the text (row 157). Additionally, ensure consistent use of the same abbreviation throughout the manuscript, whether it is PTV or PtV (e.g., rows 157, 389, and 390).

Due to these issues, the manuscript is difficult to follow.

I suggest adopting a clearer approach by including a brief description of all terms and abbreviations in the beginning of Materials and Methods section.

Please correct “alle” term row 101

The inclusion criteria were based on Little's Irregularity Index, with the authors stating they included cases of mild to moderate mandibular crowding. However, the reported range (≥ 4 mm and ≤ 9 mm) does not align with the defined classifications (Little RM. The irregularity index: a quantitative score of mandibular anterior alignment. Am J Orthod. 1975 Nov;68(5):554-63. doi: 10.1016/0002-9416(75)90086-x.):

    • 1–3 mm - mild irregularity
    • 4–6 mm - moderate irregularity
    • 7 mm or more - severe irregularity
    • 10 mm – very severe irregularity

Based on this classification, the authors have included cases of moderate and severe irregularity, not mild and moderate. This discrepancy needs to be addressed and corrected throughout the manuscript, including the abstract.

Please revise Table 1. As I understand from the text, the study included n = 70 females and n = 70 males, with 35 in each group (extraction and non-extraction). However, Table 1 indicates a total of 70 in each group, which needs to be corrected for consistency.

Table 1 is missing a title. Additionally, please move the abbreviations to appear below the table

Please revise the legend of Figure 3. The centers of resistance of the lower incisors are not displayed—please remove this reference from the legend.

Please define UK1-NB segment – row 170.

Results

Please move the abbreviations for Table 2 to appear below the table. Apply the same adjustment for Table 3 and 5 a and b.

Discussion

Please highlight the novelty of the study. Additionally, address its limitations and discuss them in light of the number of included participants and the results from the power analysis.

Author Response

Comments:

References: Please revise all references in the manuscript to ensure they are accurate and adhere to the journal’s specified format.

Done, corrected according to the the journal’s specified format

Material and Methods:
First of all, some information is assumed to be known by the reader, and abbreviations are used without prior explanation. For example, PTV plane—please define it when it first appears in the text (row 157). Additionally, ensure consistent use of the same abbreviation throughout the manuscript, whether it is PTV or PtV (e.g., rows 157, 389, and 390).

Due to these issues, the manuscript is difficult to follow.

Done, if neccessary, all abbreviation are consistently changed throughout the complete manuscript, tables and captions

I suggest adopting a clearer approach by including a brief description of all terms and abbreviations in the beginning of Materials and Methods section.

Done

Please correct “alle” term row 101

Done

The inclusion criteria were based on Little's Irregularity Index, with the authors stating they included cases of mild to moderate mandibular crowding. However, the reported range (≥ 4 mm and ≤ 9 mm) does not align with the defined classifications (Little RM. The irregularity index: a quantitative score of mandibular anterior alignment. Am J Orthod. 1975 Nov;68(5):554-63. doi: 10.1016/0002-9416(75)90086-x.):

  • 1–3 mm - mild irregularity
  • 4–6 mm - moderate irregularity
  • 7 mm or more - severe irregularity
  • 10 mm – very severe irregularity

Based on this classification, the authors have included cases of moderate and severe irregularity, not mild and moderate. This discrepancy needs to be addressed and corrected throughout the manuscript, including the abstract.

Done, the range of crowding of the included study cases has been aligned  with the defined classifications of Little

Please revise Table 1. As I understand from the text, the study included n = 70 females and n = 70 males, with 35 in each group (extraction and non-extraction). However, Table 1 indicates a total of 70 in each group, which needs to be corrected for consistency.

Table 1 is missing a title. Additionally, please move the abbreviations to appear below the table

Done, table 1 has been corrected including adding a title

Please revise the legend of Figure 3. The centers of resistance of the lower incisors are not displayed—please remove this reference from the legend.

Please define UK1-NB segment – row 170.

UK1-NB has been changed to L1-NB and defined

Results

Please move the abbreviations for Table 2 to appear below the table. Apply the same adjustment for Table 3 and 5 a and b.

Done

Discussion

Please highlight the novelty of the study. Additionally, address its limitations and discuss them in light of the number of included participants and the results from the power analysis.

Done, the novelty and the drawbacks of the study have been supplemented at the end oft he discussion section

Reviewer 2 Report

Comments and Suggestions for Authors

Dear Authors, I have to say that I had many difficulties in reading your work.

The topic covered seems to be interesting but I think the paper should to be completely revised.

I invite you to rewrite your article in a fluent English so as to make it easier to evaluate.

Do a good job.

Comments on the Quality of English Language

The quality of the English language is low.

Author Response

Comments:

The topic covered seems to be interesting but I think the paper should to be completely revised.

I invite you to rewrite your article in a fluent English so as to make it easier to evaluate.

Do a good job.

The entire manuscript was subjected to an extensive language overwork by a native speaker, to make it more readable

Reviewer 3 Report

Comments and Suggestions for Authors

Dear authors,

thank you for the opportunity to revise this manuscript of great interest.

I recommend the following revisions.

Abstract

Please reduce methods subsection with the main findings.

Introduction

-“ In borderline cases, however, analog to the upper jaw 11-12,13-15 further decision-making criteria, such as the position of molars, premolars and incisors, should be used as a basis for weighing potential consequences of nonextraction vs. extraction therapy.": please better explain this aspect in practical terms.

-“The aim of this study was to provide additional decision criteria for or against extrac- tion treatment in the mandible in borderline cases”: I would move it at the end of the section, please rephrase also the rationale for the study

M&M

“The exclusion criteria were as follows: maxillofacial surgery; previous stripping; extraction of teeth other than premo- lars; nonapplication; periodontal disease; systemic disease; and LCs with geometric aber- rations or motion artefacts.”: please explain periodontal disease classification (The latest?) and systemic disease classification (ASA?)

please divide all section in subparagraphs, to make it more readable

Results

Please reduce number of Tables and express the main findings; you can also put other materials as supplementary

Discussion

Please discuss strength and limitations of the study in terms of clinical practice.

Author Response

Comments:

Abstract

Please reduce methods subsection with the main findings.

Done, the abstract has been shortened

Introduction

-“ In borderline cases, however, analog to the upper jaw 11-12,13-15 further decision-making criteria, such as the position of molars, premolars and incisors, should be used as a basis for weighing potential consequences of nonextraction vs. extraction therapy.": please better explain this aspect in practical terms.

-“The aim of this study was to provide additional decision criteria for or against extrac- tion treatment in the mandible in borderline cases”: I would move it at the end of the section, please rephrase also the rationale for the study

Done

M&M

“The exclusion criteria were as follows: maxillofacial surgery; previous stripping; extraction of teeth other than premo- lars; nonapplication; periodontal disease; systemic disease; and LCs with geometric aber- rations or motion artefacts.”: please explain periodontal disease classification (The latest?) and systemic disease classification (ASA?)

please divide all section in subparagraphs, to make it more readable

Done, the methods section has been subdevided

Results

Please reduce number of Tables and express the main findings; you can also put other materials as supplementary

Done, Tables 4 and 5a+b have been put as supplementary materials

Discussion

Please discuss strength and limitations of the study in terms of clinical practice.

Done, this points have been added at the end of the discussion section

Round 2

Reviewer 1 Report

Comments and Suggestions for Authors

The authors have corrected the issues and responded to my requests. However, they have not used the corresponding template for the manuscript, making the revision of the revised version very difficult. Moreover, the abstract also needs to be corrected, as it does not follow the journal's template.

Author Response

The authors have corrected the issues and responded to my requests. However, they have not used the corresponding template for the manuscript, making the revision of the revised version very difficult. Moreover, the abstract also needs to be corrected, as it does not follow the journal's template.

Please revise all references in the manuscript to ensure they are accurate and adhere to the journal’s specified format.

Done, references have been corrected according to the the journal’s specified format as well as the manuscript has been transferred to the corresponding template. Moreover, the abstract has been corrected, so that it now follows the journal's template.

Reviewer 2 Report

Comments and Suggestions for Authors

However ambitious the title “Decision making in terms of treatment of crowding in the lower jaw: Extraction vs. Nonextraction ...”  is, the topic covered is certainly of interest.

As requested, the article has been rewritten in such a way that it is certainly clearer than the previous version.

However, we recommend that you delete the barred text in order to revise the grammar and punctuation at some point easily.

The Introduction section appears to be rather unorganized. Please revise.

We recommend that you transfer the beginning of the Results section to the Introduction where it fits best.

In the Results section, only search results should be entered.

In the Discussion section delete: “Clinical Relevance and Implications”, “Future Research Directions” and so on.

Nothing to say about statistical analysis.

I subscibe that “the RaV serves as an anatomical boundary for the space that defines the available space for mandibular dentition growth in the retromolar region”,  but absolutely nothing more.

Author Response

However ambitious the title “Decision making in terms of treatment of crowding in the lower jaw: Extraction vs. Nonextraction ...”  is, the topic covered is certainly of interest. As requested, the article has been rewritten in such a way that it is certainly clearer than the previous version.However, we recommend that you delete the barred text in order to revise the grammar and punctuation at some point easily.

Done

The Introduction section appears to be rather unorganized. Please revise.

Done, introduction section has been reorganized

We recommend that you transfer the beginning of the Results section to the Introduction where it fits best.

In the Results section, only search results should be entered.

Done, the results section has been reduced according to search results. In addition, the beginning of the results section has been shifted into the introduction

In the Discussion section delete: “Clinical Relevance and Implications”, “Future Research Directions” and so on.

Done

Nothing to say about statistical analysis.

I subscibe that “the RaV serves as an anatomical boundary for the space that defines the available space for mandibular dentition growth in the retromolar region”,  but absolutely nothing more.

Done, further statements about the RaV have been deleted, so that the main massage of the manuscript concentrates on the meaning of an anatomical boundary. In the discussion section headings have been removed, as requested